Heavy metal levels and human health risk implications associated with fish consumption from the lower Omo river (Lotic) and Omo delta lake (Lentic), Ethiopia

Kotacho Abiy Andemo 1 abiyandemo@hu.edu.et
Yimer Girma Tilahun 2
Sota Solomon Sorsa 2
http://orcid.org/0000-0002-9266-1126 Berego Yohannes Seifu 3
1 Biology, Hawassa University , City, Sidama , Ethiopia
2 Biology, Hawassa University , Hawassa, Sidama , Ethiopia
3 Environmental Health, Hawassa University , Hawassa, Sidama , Ethiopia
Oehlmann Jörg
Electronic publication date: 2024 Apr 29
Publication date: 2024
Volume: 12
Electronic Location ID: e17216
Received 2023 Oct 24; Accepted 2024 Mar 19
Copyright: © 2024 Kotacho et al.
Copyright year: 2024
Copyright holder: Kotacho et al.
License: This is an open access article distributed under the terms of the Creative Commons Attribution License, which permits unrestricted use, distribution, reproduction and adaptation in any medium and for any purpose provided that it is properly attributed. For attribution, the original author(s), title, publication source (PeerJ) and either DOI or URL of the article must be cited.
License URL: https://creativecommons.org/licenses/by/4.0/

Keywords: Heavy metals, Human health risk, Omo delta, Omo river

Funding: The authors received no funding for this work.

==============================
This study is the first to determine the levels of heavy metals in commercially important fish species, namely Lates niloticus and Oreochromis niloticus and the potential human health risks associated with their consumption. A total of 120 fish samples were collected from the lower Omo river and Omo delta, with 60 samples from each water source. The fish tissue samples (liver and muscle) were analyzed using a flame atomic absorption spectrometer for nine heavy metals (Cd, Co, Cr, Cu, Fe, Mn, Ni, Pb, and Zn). The human health risk assessment tools used were the target hazard quotient (THQ), the hazard index (HI), and the target cancer risk (TCR). The mean levels of heavy metals detected in the liver and muscle of L. niloticus from the lower Omo river generally occurred in the order Fe > Zn > Pb> Cu > Mn> Cr > Co > Ni and Pb > Cu > Mn > Co > Ni, respectively. The mean levels of metals in the muscle and liver tissues of O. niloticus were in the order Fe > Pb > Zn > Mn > Cu > Cr > Co > Ni and Pb > Zn > Mn > Fe > Cu > Co > Ni, respectively. Similarly, the mean levels of heavy metals detected in the liver and muscle of L. niloticus from Omo delta occurred in the order Fe > Zn > Pb > Cu > Mn > Cr > Co > Ni and Fe > Pb > Zn > Mn > Cu > Co > Cr > Ni, respectively. The mean levels in the muscle and liver tissues of O. niloticus from the Omo delta were in the order Fe > Pb > Zn > Mn > Cu > Cr > Co > Ni and Pb > Fe > Zn > Mn > Co > Cu > Ni, respectively. The study revealed that the THQ values were below 1, indicating that consumption of L. niloticus and O. niloticus from the studied sites does not pose a potential non-carcinogenic health risk. Although the TCR values for Pb in this study were within the tolerable range, it’s mean concentration in the muscle and liver tissues of both fish species from the two water bodies exceeded the permissible limit established by FAO/WHO. This is a warning sign for early intervention, and it emphasizes the need for regular monitoring of freshwater fish. Therefore, it is imperative to investigate the pollution levels and human health risks of heavy metals in fish tissues from lower Omo river and Omo delta for environmental and public health concerns.

Introduction

Aquatic products including fish are becoming increasingly popular as a source of protein, omega-3 fatty acids, vitamins, selenium, and calcium for human consumption (Kalantzi et al., 2016). The American Heart Association recommends consuming two servings of fish per week as part of a balanced diet (Neff et al., 2014). However, it is worth noting that aquatic products, due to their high fat and protein content, may also contain contaminants, which can have negative effects on human health (Usydus et al., 2009).

Oreochromis niloticus (O. niloticus) and Lates niloticus (L. niloticus) are the most commercially important fish species in Ethiopia. L. niloticus sampled from lower Omo river and Omo delta has a standard length of 45 cm and total length of 180 cm. It has terminal mouth with villi form teeth; dorsal fin long, deeply notched into anterior and posterior regions, whereas O. niloticus sampled from lower Omo river and Omo delta has a total length of 33 cm which has a mouth terminal with bicuspid teeth on the outer jaws; dark vertical bands on flank; scales between pelvic and pectoral fins distinctly smaller than those on the rest of the body; dark body; blackish opercular spot (Wakjira & Getahun, 2017).

Various natural and human caused factors, such as sewage discharge from homes or industrial, storm runoff, leaching from landfills/dumpsites, and atmospheric deposits, can cause heavy metals to accumulate in aquatic environments (Rahman, Islam & Khan, 2016). Heavy metals are significant pollutants in freshwater ecosystems and food supplies (Manal, Mahmoud & Abdel, 2014) and can pose sever risks to both humans and aquatic life (Sorsa, Gezahagn & Dadebo, 2016). The risk of consuming heavy metals from contaminated food is increasing in developing countries like Ethiopia (Berehanu, Lemma & Tekle-Giorgis, 2015; Samuel et al., 2020). Fish muscles that have accumulated heavy metals can be consumed by humans (Lubna et al., 2015; Gure, Kedir & Abduro, 2019), which can pose health risks to various vital organs such as the kidney (Manal, Mahmoud & Abdel, 2014), liver, brain (Safiur Rahman et al., 2012), lung, heart (Mir et al., 2021), and reproductive system at the cellular, tissue, and organ levels (Javed & Usmani, 2011). Excessive heavy metals in fish tissues can negatively impact early development, growth, behavior, and reproduction and damage the neurological systems of fish species (Taslima et al., 2022). Thus, the accumulation of heavy metals in human diets, including fish, is an urgent global treat that requires attention, especially in developing countries like Ethiopia.

In recent years, human activities such as agriculture, industrial, and economic development have affected the lower Omo river (Lotic) and the Omo delta (Lentic) (United States Enviromental Protection Agency (USEPA), 2010; Wakjira & Getahun, 2017). These activities have threatened the quality of the water bodies (Ojwang et al., 2010; Avery, 2012) and have been linked to the presence of heavy metals in fish tissues. In addition to extensive agricultural processing, manufacturing, and agrochemical-based irrigation projects the lower Omo river (Lotic) and Omo delta (Lentic) are subjected to pollution from urbanization and shoreline settlement (Avery, 2012). Studies have also shown that fish tissues from Lake Turkana, which is located close to the Omo delta, contain higher levels of heavy metals (Magu et al., 2016; Christof et al., 2017).

As far as we know, no research has been conducted to determine the level of heavy metals in fish tissues and its associated health risks to humans in the lower Omo river (Lotic) and the Omo delta (Lentic). However, some studies have been performed on the concentrationsof elements in fish tissue from Lake Turkana on the Kenyan side (Magu et al., 2016; Christof et al., 2017), which is the point where the southernmost tip of the Omo river extends into it. Therefore, it is crucial to determine the concentrations of heavy metals in fish tissues from the lower Omo river (Lotic) and the Omo delta (Lentic) for environmental and public health concerns. The current study aims to evaluate the human health risks associated with heavy metals present in commonly consumed fish spices (L. niloticus and O. niloticus) collected from the lower Omo river and the Omo delta. The sample were tested for the presence of nine heavy metals (Cd, Co, Cr, Cu, Fe, Mn, Ni, Pb, and Zn), and the non-carcinogenic and carcinogenic health risks to both adults and children associated with consuming fish were calculated.

Materials and Methods

Description of the study area

The Omo river basin (Fig. 1), located in southern Ethiopia, is one of the countries’ most important river systems, covering an area of approximately 79,000 km2 (Central Statistics Agency (CSA), 2017). It starts at an altitude of 2,200 m above sea level (a.s.l) and flows through the Eastern Arm of the Great Rift Valley of East Africa before finally ending in Lake Turkana at an altitude of 365 m above sea level (Wakjira & Getahun, 2017).

Figure 1 Sampling locations along the Omo river based on GPS readings.

(A) Sampling locations along the Omo river based on GPS readings (Map credit: Google Earth; Qt designer with qgis3.32.3 custom widgets). (B) Coordinates in the Omo delta fish sampling site (Map credit: Google Earth; Qt designer with qgis3.32.3).

The Omo delta, situated in the Eastern Arm of the Great Rift Valley, is approximately 50 km from Omorate Town in the downstream direction. It forms a Bird’s Foot Delta with an area of 98 km2 and lies across the Ethiopia-Kenya border in southern Ethiopia lowlands. The Omo delta starts at an altitude of 2,200 m above sea level and flows in its lower portion at an altitude that ends in Lake Turkana (Wakjira & Getahun, 2017). The Omo delta is almost 820 km South of Addis Ababa, the capital city, and 508 km from the regional city of Hawassa.

Fish sample collection and storage

Thirty samples of fish from each species, namely L. niloticus and O. niloticus, were collected from the Omo delta lake and the river water, respectively. The sampling and storage of these samples strictly followed the guidelines provided by APHA and EMERGE procedures (Rosseland et al., 2001; American Public Health Association (APHA), 2017). The fish samples were collected from fishermen who used plastic nets to trap fresh L. Niloticus and O. niloticus from the sampling sites located at (4°48′13.67″N36°2′0.37″E, 4°48′10.80″N36°2′9.16″E, 4°48′4.20″N 36° 2′8.98″E, 4°47′52.59″N 36° 2′4.80″E, 4°47′38.38″N 36° 2′16.96″E) in the river water. The fish samples were washed with deionized water just before dissection of the tissues. The fish were then dissected in the field using a plastic blade to obtain liver and muscle tissue. After removal, each liver and muscle tissue samples was carefully covered with aluminum foil and sealed simultaneously in polyethylene bags. The tissues were then separately labeled based on species and tissue type. The wraped samples were cautiously placed in an icebox and immediately transported to the Arbaminch Minch University of Chemistry Laboratory after dissection and wrapping in an icebox. The samples were then preserved in a freezer at −20 °C until analysis. The field experiments carried out were approved by the research council of Hawassa University (Approval number bio/499/13).

Sample preparation and digestion

The fish tissue samples were prepared according to the guidelines of the United States Enviromental Protection Agency (USEPA) (2010). The muscle and liver tissues were separately oven dried at 60° until they reached a constant weight. The dried tissues were then crushed in to a powder using mortar and pestle. The powdered tissue samples weighing 0.5 g each were ready for digestion. Ash digestion was carried out by taking 0.5 g of muscle and liver tissue, which was subjected to a temperature of 550 °C for 4 h. After each sample was entirely turned in to ash, it was removed and cooled in desiccators. The ash samples were mixed with 10 ml of 20% HNO3 in 50 ml beakers, placed on a hot plate, and heated slowly at 120 °C for 30 min. After digestion and cooling, dilution and filtration were done using distilled water and filter paper (Whatman No. 42). Digestion was performed following analytical method protocol for atomic absorption spectrometry (Perkin, 1996).

Sample analysis

The analysis was conducted according to APHA guidelines of 2017. Fish tissue samples were tested for heavy metal content using a flame atomic absorption spectrometer (FAAS, novAA400p). Calibration curves were prepared using analytical grade standards of each target heavy metal.

Human health risk assessment

Noncarcinogenic risks

The study aimed to determine the potential health risks posed by consuming heavy metals found in fish muscle. This was done by calculating the target hazard quotient (THQ) and hazard index (HI), which help to determine the likely hood of non-carcinogenic health hazards in humans. The THQ result assess the risk posed by a single heavy metals, while HI result calculates the cumulative risk from all heavy metals found in fish muscle. Values of THQ and HI < 1 indicate that the risk of health effects is low, while values greater than 1.0 suggest that potential noncarcinogenic health hazards are likely to occur in individuals who consume fish muscle. The degree of risk increases with higher THQ values. To determine non-carcinogenic risk, the study assessed adults and children who consumed fish muscle from the lower Omo river and Omo delta, one to seven days per week. The THQ and HI were calculated using EPA guidelines (United States Enviromental Protection Agency (USEPA), 2011, 2019) using Eqs. (1) and (2),

(1) THQ=(EF×ED×IR×Cm)(RfD×WAB×AT)×10−3

(2) HI=∑THQ

where: THQ is a non-carcinogenic health risk; the average life expectancy in Ethiopia is 65 years for adults and 6 years for children. ED is the exposure duration (United States Enviromental Protection Agency (USEPA), 2005; World Health Organization (WHO), 2015); EF is the exposure frequency which is 365 days/year for people who eat fish muscle seven times a week and 52 days/year for those who eat once a week (Food and Agriculture Organization of the United Nations (FAO), 2014), IR is the average fish ingestion rate of an individual in a day (g/day/person) which is 30 g for adults and 15 g for children in Ethiopia (United States Department of Agriculture (USDA), 2000; United States Enviromental Protection Agency (USEPA), 2005). Cm is the average concentration of heavy metals in fish muscles (mg/kg dry weight). The RfD is the oral reference dose, which is the daily ingestion of a contaminant that is unlikely to cause health effects during a life time as defined by the United States Enviromental Protection Agency (USEPA) (2003) in mg kg−1/day which is 0.001 for Cd, 0.003 (Cr), and 0.03; 0.040 (Cu), 0.7, 0.020 (Ni), 0.14 (Mn), 0.0035 (Pb) and 0.30 (Zn); WAB is the average body weight equivalent to 60 kg for adults and 21 kg for children in Ethiopians (World Health Organization (WHO), 2012); AT is the average exposure time for non-carcinogens which is (ED x EF); Cm is the average concentration of heavy metals in fish muscles (mg/kg dry weight) (World Health Organization (WHO), 2012).

Carcinogenic risk (TCR)

The target carcinogenic risks (TCR) test is a method used to determine an individual’s likelihood of developing cancer over a lifetime when exposed to a potential carcinogen. The acceptable risk levels for carcinogens typically range from 10−4 to 10−6‘ (United States Enviromental Protection Agency (USEPA), 1989). The TCR is calculating using Eq. (3).

(3) TCR=(EF×ED×Cm×IR×CPSO)×10−3(WAB×AT)

where: TCR stands for target cancer risk; CPSO represents an oral carcinogenic slope factor measured in mg/kg/day, with specific values assigned to certain substances, which is 1.7 mg kg−1/day for Ni, and 0.5 (Cr), 0.001(Cd) and 0.0085 for Pb (United States Enviromental Protection Agency (USEPA), 2010). The remaining parameters are presented in Eqs. (1) and (2).

Data quality

To insure the accuracy of our method and the validity of our results, we conducted a recovery test (American Public Health Association (APHA), 2017). This involved adding known quantities of heavy metals to fish samples, which were then digested in triplicate using the same method for the original samples. We then calculated the percent recovery using Eq. (4).

(4) Recovery=(Spikedresult−Unspikedresult)(Amountadded)×100%

All recovery values were within the acceptable range (80−120%) for heavy metal analysis (Harvey, 2000), as summarized in Fig. 2.

Figure 2 Percentage recovery in muscle of L. niloticus and O.niloticus.

Data analysis

The data were analysed using IBM SPSS 21 statistical software. We checked the normality and homogeneity of variance for data from water bodies using a Kolmogorov-Smirnov and Levene’s tests, respectively, and the data confirm that the water bodies’ data has not compromised the assumptions. Hence, we used a nonparametric test (Mann-Whitney’s U-test) to determine the differences in the levels of heavy metals between fish tissues and species. Additionally, to find the correlation between the effects of one metal concentration on the concentration of the other metal in the water bodies samples, we used Pearson correlation matrices with correlation coefficient (r) for the samples. Finally, we compare the results with the values in the literature and FAO/WHO standard limits.

Results

Levels of heavy metals in the O. niloticus in Omo river and Omo delta

Table 1 shows the average levels of different metals found in the muscle and liver tissues of O. niloticus from the Omo river and Omo delta. The mean levels of detected elements in the liver and muscle tissues of O. niloticus from the Omo river were in the following order: Fe > Pb > Zn > Mn > Cu > Cr > Co > Ni and Pb > Zn > Mn > Fe > Cu.> Co > Ni. However, in the Omo delta, the mean concentrations of detected elements in the liver and muscle tissues of O. niloticus were in the following order: Fe > Pb > Zn > Mn > Cu > Cr > Co > Ni and Pb > Fe > Zn > Mn > Co > Cu > Ni respectively. O. niloticus muscle and liver samples from the Omo river and the Omo delta both contained no detceted amount of cadmium. Additionally the order of the detceted elements in the liver of O. niloticus inhabiting both water bodies followed a similar pattern.

Table 1 Mean concentration of each HMS in liver and muscle of O. niloticus in Omo river and Omo delta.

	Omo river	Omo delta	MPL	
Liver	Muscle	Liver	Muscle	
Mean	Std. dev	Mean	Std. dev	Mean	Std. dev	Mean	Std. dev	
Mn	0.356	0.005	0.379a	0.003	0.387	0.004	0.384a	0.006	1.0	
Zn	0.477	0.424	0.424a	1.017	0.556	0.506	0.394a	0.26	40	
Cu	0.189	0.283	0.129a	0.236	0.291	0.424	0.071a	0.198	3.0	
Cr	0.126	0.075	ND		0.151	0.028	ND		0.15	
Cd	ND		ND		ND		ND		0.2	
Pb	0.908	0.210	0.790a	0.173	0.845	0.269	0.597a	0.153	0.5	
Fe	1.100	0.354	0.268a	0.059	1.741	0.691	0.411b	0.131	100	
Ni	0.014	0.002	0.010a	0.002	0.033	0.010	0.013a	0.006	0.15	
Co	0.068	0.020	0.054a	0.017	0.065	0.029	0.080b	0.023	–	
Note:

Mean concentration of each heavy metals for having different letter in rows are statistically different. ND, not detected; MPL, maximum permissible limit in human diet according to FAO/WHO (1989).

The study found that the mean zinc concentration in the muscle tissue of O. niloticus was 0.424 and 0.394 mg kg−1 in Omo river and Omo delta, respectively (Table 1). The level of Zinc in the liver tissue of O. niloticus was 0.556 mg kg−1 in the Omo delta and 0.477 mg kg−1 in the Omo river. The mean concentration of Copper (Cu) in the liver tissues of O. niloticus were 0.189 and 0.291 mg kg−1 in the Omo river and Omo delta respectively.

The mean concentrations of manganese (Mn) were 0.356 mg kg−1 in the liver and 0.379 mg kg−1 in the muscle of O. niloticus. However, in O.niloticus the concentrations of Mn were 0.387 and 0.384 mg kg−1 in the liver and muscle respectively.

The average Cr content in O.niloticus tissue is displayed in Table 1. The average Cr concentration in O. niloticus livers from the Omo river was 0.145 mg kg−1, which was less than the average Cr concentration in the Omo delta, which was 0.154 mg kg−1. Cr was not detected in the muscle tissues of O.niloticus in the Omo river and Omo delta. O. niloticus muscle and liver samples from the Omo river and the Omo delta both contained no detected amount of cadmium. The level of lead (Pb) in the muscle tissues ranged from 0.790 to 0.597 mg kg−1. The sample from Omo river had a greater value, but the mean concentration in the muscle tissue was not significantly different (p.value > 0.05). The analysis revealed that with the exception of iron and cobalt, the mean levels of all detected heavy metals were not significantly different (p.value > 0.05) in the muscle tissue of O. niloticus between the Omo river and Omo delta (Table 1).

The study found that the mean iron concentration in the muscle tissues of O. niloticus was 0.268 mg kg−1 in the Omo river and 0.411 mg kg−1 in the Omo delta. The findings of the present study showed that the Fe concentration in the liver of O. niloticus was 1.100 mg kg−1 in the Omo river and 1.74 mg kg−1 in the Omo delta. The results of the study also showed a significant difference (p value < 0.01) in the average iron concentration between Omo river and Omo delta samples.

The concentration of Cobalt and nickel in the muscle and liver tissue of O. niloticus varied in the Omo river and Omo delta. The mean nickel concentrations of O. niloticus in the muscle tissues were 0.010 and 0.013 mg kg−1 in the Omo river and Omo delta, respectively as shown in Table 1. On the other hand, the mean Ni concentrations in the liver tissue of O. niloticus in the Omo river and Omo delta were 0.014 and 0.018 mg kg−1, respectively. These findings indicate that the concentration of Ni in the liver tissue of O. niloticus was higher than that in the muscle tissue. The result also showed that Cobalt concentrations in the muscle tissue of O. niloticus ranged from 0.054 to 0.080 mg kg−1 in the Omo river and Omo delta respectively. There was statistically significant variation in the mean Co concentration in the muscle tissues of O. niloticus between the two sites (Omo river and Omo delta) with a p value < 0.01.

Table 2 presents the correlation among heavy metals in the fish muscles of O. niloticus and L. niloticus from the lower Omo river and Omo delta. The study analyzed the correlations among heavy metals in fish tissues. The results of Pearson’s correlation coefficients revealed some significant (p < 0.01, p < 0.05) correlations between heavy metals in the muscle of O. niloticus from the lower Omo river. Specifically, there were significant correlation between Cr and Fe (r = 0.753), and Cr and Ni (r = (0.702), Fe and Ni (r = 0.65), Co and Fe (r = 0.482),and Fe and Zn (r = −0.23)). Similarly, A significant correlation was observed in the muscle of O. niloticus from Omo delta between Cr and Fe (r = 0.705), Cr and Ni (r = 0.683), Cr and Pb (r = 0.533), Pb and Fe (r = 0.480), Pb and Cu (r = 0.450), Co and Cu (r = −0.381) (Table 2).

Table 2 T test in the muscle of O. niloticus in Omo river and Omo delta.

		Mn	Zn	Cu	Cr	Cd	Pb	Fe	Ni	Co	
Omo river	Mn	1.00									
Zn	−0.10	1.00								
Cu	0.11	−0.07	1.00							
Cr	0.25	−0.19	0.27	1.00						
Cd	0.437**	−0.416*	0.414*	0.426**	1.00					
Pd	−0.03	−0.15	−0.17	0.24	−0.08	1.00				
Fe	0.22	−0.23	0.32	0.753**	0.554**	0.06	1.00			
Ni	0.07	−0.08	0.31	0.702**	0.572**	0.11	0.650**	1.00		
Co	0.13	0.21	0.12	0.336*	−0.14	0.07	0.482**	0.14	1.00	
Omo delta	Mn	1.0									
Zn	−0.3	1.0								
Cu	0.535**	−0.2	1.0							
Cr	0.2	−0.2	0.388*	1.0						
Cd	0.1	−0.2	0.345*	0.523**	1.0					
Pd	0.3	−0.1	0.450**	0.533**	0.535**	1.0				
Fe	0.3	−0.1	0.442**	0.705**	0.335*	0.489**	1.0			
Ni	0.2	−0.2	0.2	0.638**	0.1	0.3	0.571**	1.0		
Co	0.0	0.0	−0.381*	−0.3	−0.460**	−0.2	−0.2	0.1	1.0	
Notes:

* Significant at the 0.05 level (2-tailed).

** Significant at the 0.01 level.

Levels of heavy metals in L. niloticus in the Omo river and Omo delta

As shown in Table 3, the highest concentration of heavy metals detected in the liver of L. niloticus was Fe (2.918 ± 1.47 mg kg−1) and the lowest mean concentration was Ni (0.011 ± 0.003 mg kg−1). Overall, the levels of metals in the liver and muscle of L. niloticus followed the order Fe > Zn > Pb > Cu > Mn > Cr > Co > Ni and Pb > Cu > Mn > Cr > Co > Ni respectively. L. niloticus muscle and liver samples from the Omo river and the Omo delta both contained no detected amount of cadmium.

Table 3 Mean concentration of each HMS in liver and muscle of L. niloticus in Omo river and Omo delta.

	Omo river	Omo delta	MPL	
Liver	Muscle	Liver	Muscle	
Mean	Std. dev	Mean	Std. dev	Mean	Std. dev	Mean	Std. dev	
Mn	0.391	0.003	0.383a	0.003	0.394	0.004	0.385a	0.005	1.0	
Zn	1.01	0.482	0.642a	0.474	1.127	0.870	0.428a	0.393	40	
Cu	0.481	0.459	0.157a	0.273	0.407	0.419	0.13a	0.283	3.0	
Cr	0.145	0.021	ND		0.154	0.023	0.039	0.085	0.15	
Cd	ND		ND		ND		ND		0.2	
Pd	1.009	0.215	0.793a	0.173	1.124	0.151	0.89b	0.099	0.5	
Fe	2.918	1.476	0.509a	0.273	1.810	0.465	0.940b	0.395	100	
Ni	0.019	0.006	0.011a	0.003	0.018	0.011	0.019b	0.003	0.15	
Co	0.097	0.023	0.080a	0.028	0.085	0.015	0.045b	0.021	–	
Note:

Mean concentration of each heavy metals for having different letter in rows are statistically different. MPL, maximum permissible limit in human diet according to FAO/WHO (1989).

The concentrations of zinc in the muscle tissue of L. niloticus were 0.642 and 0.428 mg kg−1 in Omo river and Omo delta respectively. The copper (Cu) concentration in L. niloticus muscle tissues from the Omo river and Omo delta varied between 0.157 and 0.13 mg kg−1, respectively. The copper concentration in L. niloticus muscle tissues from the Omo river and Omo delta varied between 0.157 and 0.13 mg kg−1, respectively. However, the mean copper concentration in liver tissues of L. niloticus ranged from 0.481 to 0.407 mg kg−1 in the Omo river and Omo delta, respectively. The levels of manganese in the muscle tissue of L. niloticus were 0.383 and 0.385 mg kg−1 in the Omo river and Omo delta, respectively.

The concentration of chromium (Cr) was found to be below the detection limit up to 0.154 mg kg−1. The liver tissues of L. niloticus showed a higher level of chromium, exhibited significant site dependent variation in the mean Cr concentration (p value < 0.02). The lead (Pb) level in L. niloticus muscle tissues varied between 0.793 and 0.890 mg kg−1 in the Omo river and Omo delta respectively. As presented in Table 3, the mean Pb concentration in L. niloticus muscle tissue significantly differed by site (p value < 0.01) (Omo river and Omo delta).

The mean concentrations of Fe in the muscle tissue of L. niloticus were 0.509 and 0.94 mg kg−1 in the Omo river and Omo delta, respectively. In the Omo river and Omo delta, the mean Fe levels in L. niloticus muscle tissue differed significantly (p value < 0.01) (Table 3). The mean nickel (Ni) concentration ranged from 0.011 to 0.019 kg−1 in the Omo river and Omo delta, respectively. Similarly, the mean nickel concentration in L. niloticus muscle tissue significantly differed (p value < 0.01) depending on location (Omo river and Omo delta).

Table 4 shows the results of Pearson’s correlation coefficients in the muscle of L.niloticus from Omo delta. The analysis revealed significant correlation between various metals, such as Cr and Ni (r = 0.623), Cr and Fe (r = 0.622), Cr and Mn (r = 0.604), Pb and Cu (r = 0.526), Ni and Mn (r = 0.519), Pb and Cr (r = 0.494). Additionally, there was a significant correlation in L. niloticus between Cr and Fe (r = 0.671), Co and Fe (r = 0.545), Pb and Co (r = 0.517), Pb and Fe (r = 529), Pb and Ni (r = −0.454), and Cr and Zn (r = −0.3810) (Table 4).

Table 4 T test in the muscle of L. niloticus Omo river and Omo delta.

		Mn	Zn	Cu	Cr	Cd	Pd	Fe	Ni	Co	
Omo river	Mn	1.0									
Zn	−0.2	1.0								
Cu	0.321*	−0.360*	1.0							
Cr	0.604**	−0.310*	0.329*	1.0						
Cd	0.2	0.0	0.460**	0.411**	1.0					
Pd	0.476**	−0.2	0.526**	0.494**	0.620**	1.0				
Fe	0.466**	−0.3	0.373*	0.622**	0.466**	0.517**	1.0			
Ni	0.519**	−0.2	0.376*	0.623**	0.505**	0.336*	0.466**	1.0		
Co	−0.2	−0.2	0.2	0.0	0.2	0.0	0.2	−0.1	1.0	
Omo delta	Mn	1									
Zn	−0.072	1								
Cu	0.122	0.018	1							
Cr	0.116	−0.381*	−0.069	1						
Cd	−0.053	0.500**	0.264	−0.392*	1					
Pd	0.15	−0.339	0.107	0.373*	−0.459*	1				
Fe	0.226	−0.281	0.31	0.671**	−0.277	0.529**	1			
Ni	0.114	−0.135	−0.102	−0.023	−0.04	−0.454*	−0.142	1		
Co	0.076	−0.291	0.214	0.411*	−0.623**	0.517**	0.545**	−0.159	1	
Notes:

* Significant at the 0.05 level (2-tailed).

** Significant at the 0.01 level.

Table 5 presents the difference in the mean level of heavy metals found in fish tissues and species from Omo delta. The mean level of heavy metals in the liver and muscle of L.niloticus from Omo delta were significantly different for Zn, Cr, Pb, Fe, and Co. A similar difference was observed in O.niloticus for all detected heavy metals except for Zn and Co. There was a difference in the heavy metal levels based on fish species observed between the liver of O.niloticus and L.niloticus for Pb, Ni, and Co. Furthermore, a significant difference was observed between the mean heavy metal levels in the muscle of O.niloticus and L. niloticus for Pb, Fe, and Ni. The differences in the mean level of heavy metals among the fish’s tissues and species from lower Omo river are also presented in (Table 5).

Table 5 T-test among tissues of fish species.

	Tissues o fish species	Sig. (two-tailed)	
		Mn	Zn	Cu	Cr	Pb	Fe	Ni	Co	
Omo delta	liver and muscle of L. niloticus	0.4	0.02*	0.23	0.00**	0.00**	0.00**	0.83	0.00**	
liver and muscle of O. niloticus	0.05*	0.24	0.00**	0.00**	0.00**	0.00**	0.00**	0.11	
liver of O. niloticus and L. niloticus	0.806	0.775	0.464	0.769	0.001**	0.762	0.001**	0.041**	
Omo river	liver and muscle of L. niloticus	0.00**	0.02**	0.01**	0.00**	0.00**	0.00**	0.00**	0.88	
liver and muscle of O. niloticus	0.934	0.242	0.496	0.000**	0.075**	0.000**	0.00**	0.030**	
liver of of O. niloticus and L. niloticus	0.504	0.251	0.0227**	0.241	0.135	0.000**	0.0023**	0.0551*	
Muscle of O. niloticus and L. niloticus	0.002**	0.394	0.998	0.074*	0.969	0.001**	0.296	0.002**	
Notes:

* Significant at the 0.05 level (2-tailed).

** Significant at the 0.01 level.

Human health risk of heavy metals through consumption of fish from the Omo river

The potential non-carcinogenic health risks associated with consuming muscle tissue from L. niloticus and O. niloticus fish species from the lower Omo river were assessed by calculating the target hazard quotient (THQ) and the Hazard Index (HI). Table 6 shows THQ and HI indices for individuals who consumed fish muscle tissues ranging from once to seven times a week. The THQs for the heavy metals in the muscle of L. niloticus and O. niloticus are ranked in the following order: Pb > Cu > Mn > Co > Zn > Fe > Ni and Pb > Cu > Mn > Co > Ni > Zn > Fe respectively (Figs. 3 and 4).

Table 6 Estimated target hazard quotient (THQ) and Hazard Index (HI) in adults (a) and children (c) due to heavy metal exposure in muscle of L. niloticus and O. niloticus.

Fish species	Level of exposure (d/w)	Target hazard quotient (THQ)
adult (a) and children (c)	Hazard index
(HI)	
		Mn	Zn	Cu	Pb	Fe	Ni	Co		
L. niloticus	1	0.00096a	0.00072a	0.00096a	0.0768a	0.00024a	0.000192a	0.000912a	0.080784	
	0.00144c	0.001104c	0.002064c	0.096c	0.00096c	0.000288c	0.00144c	0.103296	
2	0.001872a	0.001488a	0.002688a	0.1536a	0.00048a	0.000384a	0.001824a	0.162336	
	0.00288c	0.0024c	0.00432c	0.24c	0.001632c	0.000576c	0.00288c	0.254688	
3	0.002784a	0.002208a	0.004032a	0.2304a	0.00072a	0.000576a	0.002736a	0.243456	
	0.00432c	0.00336c	0.00624c	0.336c	0.0024c	0.00096c	0.00432c	0.3576	
4	0.003744a	0.002928a	0.00528a	0.312a	0.001008a	0.000768a	0.003648a	0.329376	
	0.00576c	0.004464c	0.0096c	0.48c	0.00336c	0.001152c	0.00576c	0.510096	
5	0.004656a	0.003648a	0.00672a	0.3888a	0.001248a	0.00096a	0.00456a	0.410592	
	0.0072c	0.00576c	0.01056c	0.576c	0.00432c	0.00144c	0.0072c	0.61248	
6	0.00576a	0.004416a	0.00816a	0.4656a	0.001488a	0.001152a	0.00528a	0.491856	
	0.00864c	0.00672c	0.01248c	0.72c	0.0048c	0.00192c	0.00816c	0.76272	
7	0.00672a	0.00528a	0.00864a	0.528a	0.001728a	0.001344a	0.00624a	0.557952	
O. niloticus		0.01008c	0.0096c	0.0144c	0.96c	0.00576c	0.002016c	0.0096c	1.011456	
1	0.000912a	0.00048a	0.001104a	0.0768a	0.00013a	0.000192a	0.000624a	0.080242	
	0.00144c	0.00072c	0.00192c	0.12c	0.000192c	0.00024c	0.000912c	0.125424	
2	0.001824a	0.00096a	0.002208a	0.1536a	0.000264a	0.000336a	0.001248a	0.16044	
	0.00288c	0.00144c	0.00336c	0.24c	0.000398c	0.000576c	0.0024c	0.251054	
3	0.002736a	0.00144a	0.003312a	0.1536a	0.000394a	0.000528a	0.001824a	0.163834	
	0.00432c	0.0024c	0.00528c	0.336c	0.000624c	0.000768c	0.00288c	0.352272	
4	0.003696a	0.00192a	0.004416a	0.3072a	0.000528a	0.000672a	0.0024a	0.320832	
	0.00576c	0.002976c	0.00672c	0.48c	0.000816c	0.00096c	0.00384c	0.501072	
5	0.004608a	0.0024a	0.00576a	0.384a	0.000672a	0.000864a	0.003072a	0.401376	
	0.0072c	0.00384c	0.024c	0.576c	0.00096c	0.001296c	0.0048c	0.618096	
6	0.00528a	0.00288a	0.00672a	0.4608a	0.000768a	0.00096a	0.00384a	0.481248	
	0.00864c	0.00384c	0.0096c	0.72c	0.0012c	0.001584c	0.00576c	0.750624	
7	0.00624a	0.003408a	0.00768a	0.528a	0.000912a	0.003696a	0.00432a	0.554256	
	0.01008c	0.00528c	0.0144c	0.96c	0.00144c	0.00192c	0.00672c	0.99984	
Note:

Cd and Cr were not detected in the muscle of both fish species. (d/w), days/week. The bold indicates HI values greater than 1.0 suggesting that potential noncarcinogenic health hazards.

Figure 3 Target hazard quotients (THQ) of heavy metals in fish species from Omo river and Omo delta for adults.

Figure 4 Target hazard quotients (THQ) of heavy metals in fish species from Omo river and Omo delta for children.

All of the tested sample showed that THQs for heavy metals found in fish muscle that was consumed by both adult and children were less than one. However, when considering HI for the heavy metal detected, it was found that index for children was greater than one. The average contribution to the THQ for the HI showed that lead, copper, and manganese for almost 97% to the HI via the consumption of both fish specis’s muscle tissues. Pb while single-handedly contributed about 90% to HI via the muscle tissues of both fish species.

The result showed that the HI values for L. niloticus were 0.558 (for adults) and 1.01 (for children), while, the HI values for O. niloticus were 0.555 (for adults) and 0.1 (for children). Additionally, the text mentions the THQ values for various heavy metals found in the fish, including lead and nickel. The results showed that the maximum THQ and HI values were observed for Pb, while the minimum was observed in Ni (Table 6).

Table 7 shows the estimated risk of developing cancer from consuming Pb and Ni through muscle tissue of L. niloticus and O. niloticus, for a period of 1 to 7 days per weeks. Accordingly, the target cancer risk (TCR) values in the muscle tissue of both L. niloticus and O. niloticus were found to be greater for nickel as compared to lead (Table 7).

Table 7 Target cancer risk (TCR) in adults and children due to heavy metal exposure in muscle of L. niloticus and O. niloticus.

Fish
species	Level of exposure
(d/w)	Carcinogenic risk (CR) in
adults	Carcinogenic risk (CR)
children	
		Pb	Ni	Pb	Ni	
L. niloticus	1	2.30E−06	6.24E−06	2.69E−06	1.01E−05	
2	4.61E−06	1.30E−05	7.20E−06	1.97E−05	
3	7.20E−06	1.63E−05	1.06E−05	2.93E−05	
4	9.12E−06	2.54E−05	1.39E−05	3.89E−05	
5	1.15E−05	3.22E−05	1.78E−05	4.80E−05	
6	1.39E−05	3.84E−05	2.11E−05	5.76E−05	
7	1.63E−05	4.51E−05	2.45E−05	6.72E−05	
O. niloticus	1	2.30E−06	5.76E−06	3.50E−06	9.12E−06	
2	4.61E−06	1.15E−05	7.20E−06	1.78E−05	
3	6.72E−06	1.73E−05	1.06E−05	2.64E−05	
4	9.12E−06	2.30E−05	1.39E−05	3.55E−05	
5	1.10E−05	2.88E−05	2.02E−05	4.42E−05	
6	1.34E−05	3.50E−05	2.11E−05	5.28E−05	
7	1.58E−05	4.08E−05	2.45E−05	6.24E−05	

Human health risk of heavy metals through the consumption of fish from the Omo delta

Using the THQ and HI, the noncarcinogenic risks of the heavy metals identified in the muscle of L. niloticus and O. niloticus from the Omo delta were evaluated in adults (Table 8) and children (Table 9). Tables 8 and 9 for adults and children, respectively, show the index findings (THQ and HI) from eating fish muscle one to seven times a week. The THQs values in the muscle of L. niloticus and O. niloticus were in the order Pb > Cr > Cu > Mn > Co > Zn > Fe > Ni and Pb > Mn > Co > Cu > Zn > Ni > Fe respectively. The index (HI) values due to consumption of L. niloticus muscle were 0.668 (for adults) and 0.433 (for children). Similarly, the HI values in O. niloticus were 0.942 (for adults) and 0.441 (for children). The maximum THQs values were observed for Pb in both L. niloticus and O. niloticus whereas, the minimum was observed for Fe in muscle of O. niloticus and Ni in L. niloticus.

Table 8 Estimated target hazard quotient (THQ) and Hazard Index (HI) in adults due to heavy metal exposure in muscle of L. niloticus and O. niloticus.

Fish species	Level of exposure (d/w)	Target hazard quotient (THQ)	Hazard index (HI)	
		Mn	Zn	Cu	Cr	Pb	Fe	Ni	Co		
L. niloticus	1	0.000941	0.00049	0.001104	0.004445	0.086947	4.59E−04	3.25E−04	0.000514	9.55E−02	
3	0.002822	0.001464	0.003307	0.013334	0.260837	1.38E−03	9.74E−04	0.001541	2.86E−01	
5	0.004699	0.002438	0.005515	0.022224	0.434726	2.29E−03	1.62E−03	0.002563	4.76E−01	
7	0.0066	0.003422	0.007742	0.0312	0.610286	3.22E−03	2.28E−03	0.0036	6.68E−01	
O. niloticus	1	0.000936	0.000449	0.000605	ND	0.05832	2.01E−04	2.26E−04	0.000936	6.17E−02	
3	0.002813	0.001349	0.001819	ND	0.174965	6.00E−04	6.67E−04	0.002803	1.85E−01	
5	0.00469	0.002246	0.003034	ND	0.291605	1.00E−03	1.11E−03	0.004675	3.08E−01	
7	0.006581	0.003154	0.004262	ND	0.409373	1.41E−03	1.56E−03	0.006562	4.33E−01	

Table 9 Estimated target hazard quotient (THQ) and hazard index (HI) in children due to heavy metal exposure in muscle of L. niloticus and O. niloticus.

Fish species	Level of exposure (d/w)	Target hazard quotient (THQ)	Hazard index
(HI)	
		Mn	Zn	Cu	Cr	Pb	Fe	Ni	Co		
L. niloticus	1	0.001344	0.000696	0.001574	0.00635	0.124205	0.000658	4.64E−04	0.000734	1.36E−01	
3	0.004032	0.002093	0.004728	0.019051	0.372619	0.001968	1.39E−03	0.002198	4.08E−01	
5	0.006715	0.003485	0.007877	0.031747	0.621034	0.003278	2.32E−03	0.003662	6.80E−01	
7	0.009427	0.004891	0.011059	0.044573	0.871838	0.004603	3.26E−03	0.005141	9.42E−01	
O. niloticus	1	0.001339	0.000643	0.000869	ND	0.083318	0.000287	3.17E−04	0.001334	8.81E−02	
3	0.004018	0.001925	0.002602	ND	0.24995	0.000859	9.50E−04	0.004003	2.64E−01	
5	0.006701	0.003206	0.004334	ND	0.416582	0.001435	1.59E−03	0.006677	4.41E−01	
7	0.043886	0.004502	0.006086	ND	0.584818	0.002011	2.23E−03	0.00937	6.53E−01	
Note:

ND, not detected; Cd was not detected in the muscle and liver of both fish species.

The probable target cancer risk due to the ingestion of Cr, Pb, and Ni through the muscle of L. niloticus and O. niloticus for 1 to 7 days a week is presented in Table 10. The target cancer risk (TCR) values in the muscle of both L. niloticus and O. niloticus were in an order Ni > Cr > Pb (Table 10). The THQ for heavy metals in fish muscle consumed by adults and children was less than one in every sample that was examined, and the Hazard Index (HI) of the identified heavy metals was also less than one.

Table 10 Target cancer risk (TCR) in adults and children due to heavy metal exposure in muscle of L. niloticus and O. niloticus.

Fish species	Level of exposure (d/w)	Carcinogenic risk (CR)	Carcinogenic risk (CR)	
Adults	Children	
Cr	Ni	Pb	Cr	Ni	Pb	
L. niloticus	1	6.67E−06	1.10E−05	2.59E−06	9.53E−06	1.58E−05	3.70E−06	
3	2.00E−05	3.31E−05	7.76E−06	2.86E−05	4.73E−05	1.11E−05	
5	3.33E−05	5.52E−05	1.29E−05	4.76E−05	7.89E−05	1.85E−05	
7	4.68E−05	7.75E−05	1.82E−05	6.69E−05	1.11E−04	2.59E−05	
O. niloticus	1	ND	7.56E−06	1.74E−06	ND	1.08E−05	2.48E−06	
3	ND	2.27E−05	5.21E−06	ND	3.24E−05	7.44E−06	
5	ND	3.78E−05	8.67E−06	ND	5.39E−05	1.24E−05	
7	ND	5.30E−05	1.22E−05	ND	7.58E−05	1.74E−05	
Note:

d/w, days/week; E, exponent; ND, not detected. E, exponent (power of ten). Cd was not detected in the muscle of both fish species.

Discussion

The levels of lead (Pb) in muscle tissues of O. niloticus were found to range from 0.597 to 0.890 mg kg−1, with higher concentration observed in the samples taken from the Omo delta area. This could be attributed to the water quality of the Omo river and its delta regiion. These results differ from those reported by Gure, Kedir & Abduro (2019), Esilaba et al. (2020), and Fredrick et al. (2021), who found mean concentrations of lead at 8.39, 9.99, and 5.8 mg kg−1, respectively. In addition, the Pb levels in muscle tissues at both the Omo river and Omo delta sites were found to be higher than those previously reported by Magu et al. (2016) and Samuel et al. (2020). The high levels of lead coud be attributed to heavy agricultural runoff, which contain pesticides, agrochemicals, fertilizers, and petrol from fishing boats that contain lead. However, the Pb content in the liver tissue of O. niloticus in the present study was found to be lower than the levels reported by Dugasa & Endale (2018) (1.63 mg kg−1) and Gure, Kedir & Abduro (2019) (6.02 mg kg−1). The mean Pb levels in the muscle tissue of O. niloticus from both the Omo river and Omo delta exceeded the recommended limit for human consumption as set by Food and Agriculture Organization of the United Nations (FAO) (2014). These findings indicate that consuming fish muscle tissue from the Omo river and Omo delta could be harmful due to Pb toxicity.

The concentration of lead (Pb) in the muscle tisssues of L. niloticus varied between 0.793 and 0.89 mg kg−1 in the Omo river and Omo delta, respectively. Our study found that the lead levels were higher than those reported in prior studies. For instance, the Pb concentrations of 0.131 ± 0.048 and 0.181 ± 0.664 mg kg−1were reported in the River Nile (Aswan) and Nasser Lake by Al-Hossainy et al. (2017). Similarly, Machiwa (2005) reported Pb concentration in the range <0.01–0.08 mg kg−1 in Lake Victoria, Tanzania which are lower than our findings. Our study also found that the Pb levels in the muscle tissues of L. niloticus were higher than those reported by Magu et al. (2016), Lubna et al. (2015) and Samuel et al. (2020) but lower than those reports by Esilaba et al. (2020), Fredrick et al. (2021). The high lead levels in our study could be attributed to heavy rains that carried wastes down, contributed to the greater metal level in L. niloticus (Dural, Göksu & Özak, 2007; Saei-Dehkordi & Fallah, 2011). Anthropogenic activities like use of agrochemicals, Car washing, gas/fuel station, solid wastes, and effluents from factories could also account for the high lead levels. It is worth noting that the mean lead levels in L. niloticus muscle tissues were above the FAO/WHO (1989).

The findings of this study indicate that the concentration of manganese in the muscle tissue of O. niloticus 0.379 mg kg−1 in Omo river. This lower than the manganese levels (1.972 mg kg−1) reported by Haile et al. (2015) in Ethiopia and Mn concentration (0.77 mg kg−1) reported by Magna et al. (2021) in volta river basin of Ghana. However, the level is comparable to manganese concentration (0.55 mg kg−1) found in muscle tissue of O. niloticus from Lake Hawassa, as reported by Abayneh, Tadesse & Chandravanshi (2003). It is worth noting that this observed value is below the permitted human food intake level established by FAO/WHO (1989).

The levels of zinc in O. niloticus muscle tissue were 0.424 mg kg−1 in the Omo river and 0.394 mg kg−1 in the Omo delta. The concentration of Zn in the muscle of O. niloticus from the Omo river (0.424 mg kg−1) was similar to that reported by Zenebe (2011) in Ethiopia (0.26 mg kg−1). However, it was lower than that reported by Magu et al. (2016) in freshwater fish in Kenya (0.647 mg kg−1) and Lubna et al. (2015) in Langat River in Malaysia (19.36 mg kg−1). On the other hand, the concentration of Zn in O. niloticus muscle from the Omo delta was lower than that in the muscle of the same species as reported by Abayneh, Tadesse & Chandravanshi (2003), Haile et al. (2015) and Samuel et al. (2020) (4.76, 21.11, and 19.36 mg kg−1, respectively). As per standards set by FAO/WHO (1989), the mean zinc concentration in the muscle of O. niloticus from the Omo river and Omo delta was lower than the MPL (maximum permissible limit) for human diet. Therefore it can be concluded that, O. niloticus from the Omo river and Omo delta are safe for human consumption in terms of Zn toxicity.

The concentration of zinc in muscle tissue of L. niloticus was 0.642 and 0.428 mg kg−1 in the Omo river and Omo delta, respectively. The level of Zn in muscle of L. niloticus in this finding was also higher than the earlier study by Zenebe (2011). Similarly, our finding was higher than that of a previous study (Machiwa, 2005), in which the Pb concentration was reported to be in the range <0.01–0.0189 mg kg−1 in Lake Victoria, Tanzania.

The level of copper (Cu) in the muscle tissue of O. niloticus varied from 0.129 to 0.071 mg kg−1, with the Omo river exhibiting greater levels. The levels are below the maximum permitted limit in the human diet established by the FAO/WHO (1989). The study also revealed that the mean copper level of muscle tissues in O. niloticus were higher than those reported by Zenebe (2011), Cu 0.03 mg kg−1. Furthermore, the present findings were higher than the results of the previous studies on copper concentration in Lake victoria Tanzania, which ranged between 0.01–0.097 mg kg−1(Machiwa, 2005). This may be attributed to agricultural runoff, which may carry higher values of these metals and arise from anthropogenic activities such as the use of chemical fertilizers and pesticides in agriculture land. However, the finding of our study was lower than that of the previous studies, which included Cu (0.419 mg kg−1) (Magu et al., 2016), Cu (13.833 mg kg−1) (Haile et al., 2015), and Cu (4.64 mg kg−1) (Gure, Kedir & Abduro, 2019). According to FAO/WHO (1989), O. niloticus from the Omo river and Omo delta had mean Cu concentrations in their muscles that were below the MPL for humans’ diets. These findings demonstrated that O. niloticus from the Omo river and Omo delta may be safe for consumption by humans because of Cu toxicity.

The copper (Cu) level in L. niloticus muscle tissues varied between 0.157 and 0.130 mg kg−1 in the Omo river and Omo delta respectively. The levels are below the FAO/WHO (1989) maximum permissible limit in the human diet. The mean Cu levels of muscle tissues in O. niloticus of this study were higher than the earlier report by Zenebe (2011). Similarly, the present finding were higher than those of the prior studies on the Cu concentrations in the range 0.001–0.097 mg kg−1 (Machiwa, 2005) in Lake Victoria, Tanzania. The mean muscle content of Cu in L. niloticus of the current study was lower than the previous study reports by Esilaba et al. (2020) but lower than the study report by Magu et al. (2016).

The current findings showed that the mean Cu concentration in O. niloticus liver tissues was 0.407 mg kg−1 in the Omo delta and (0.481 mg kg−1) in the Omo river. These values are higher than the findings reported by Dugasa & Endale (2018) which were at 0.029 mg kg−1. However, the results are lower than those of a previous study that reported a concentration of Cu at 8.28 mg kg−1 (Gure, Kedir & Abduro, 2019).

According to the findings of the current investigation, O. niloticus tissue had a lower Cr concentration than that reported by Lubna et al. (2015). Cr = 1.48 mg kg−1; and Gure, Kedir & Abduro (2019). Cr = 10.3 mg kg−1. However, the levels of Cr in the liver tissue of O. niloticus was 0.126 mg kg−1 in the Omo river and 0.151 mg kg−1in the Omo river, which was lower than the Cr (8.28 mg kg−1) levels reported in a report by Gure, Kedir & Abduro (2019) from the Gibe River in Ethiopia.

The chromium (Cr) levels ranged from being below the detection limit to 0.154 mg kg−1. Higher level of Cr was observed in the liver tissues of L. niloticus. The muscle content of Cr in L. niloticus of this result (0.039 mg kg−1) was lower than that in previous studies (Samuel et al., 2020; Magna et al., 2021). However, the levels of Cr in the liver tissue of L. niloticus in the current study was higher than the study reported by Dugasa & Endale (2018) and lower than that in the study by Gure, Kedir & Abduro (2019) from Ethiopia.

The study found that the muscle tissues of O. niloticus from the Omo river and Omo delta had mean iron concentrations of 0.268 and 0.411 mg kg−1, respectively. These levels were lower than those reported by other studies, such as Abayneh, Tadesse & Chandravanshi, 2003 and Samuel et al. (2020), who reported 5.49 and 11.34 mg kg−1, respectively. Similarly, the current study showed lower levels of Iron in O. niloticus muscle tissues than those found in other research (Abayneh, Tadesse & Chandravanshi, 2003; Dugasa & Endale, 2018; Magna et al., 2021). Additionally the levels of Iron in muscle tissue were within the allowable limit set by FAO/WHO (2011), indicating that they are safe for human consumption. Regarding the liver tissues, the study found that Fe concentration in the liver of O. niloticus was 1.100 mg kg−1 in the Omo river and 1.74 mg kg−1 in the Omo delta, which was greater than the earlier Fe (0.809 mg kg−1) reported by Dugasa & Endale (2018).

The mean concentrations of Fe in the muscle tissue of L. niloticus were 0.509 and 0.94 mg kg−1 in the Omo river and Omo delta, respectively. The tissue levels of Fe in this study were below the FAO/WHO (2011) allowable limit. These concentrations are within the FAO/WHO (1989) recommended permissible human diet intake levels. The Ni levels of muscle tissues in L. niloticus in the present finding were comparable to those in previous reports (Zenebe, 2011; Magu et al., 2016; Samuel et al., 2020) and lower than the studies recorded by Magna et al. (2021).

The results of this investigation showed that the mean concentrations of nickel in O. niloticus muscle in the Omo river and Omo delta were 0.010 and 0.013 mg kg−1, respectively. These concentrations are within the FAO/WHO (1989) recommended permissible human diet intake levels, which cannot impose immediate adverse health effects. The Ni levels of muscle tissues from O. niloticus and L. niloticus in the present finding were comparable to those in previous reports (Zenebe, 2011; Samuel et al., 2020) and lower than those in studies (Magu et al., 2016; Magna et al., 2021). The cobalt levels O. niloticus muscle tissues varied from 0.054 to 0.082 mg kg−1 in the Omo river and Omo delta, respectively. The mean muscle level of cobalt in O. niloticus was comparable with the previous report by Samuel et al. (2020). However, the mean Co level in the current finding was lower than that in the study recorded of the muscle and liver tissue of O. niloticus by Gure, Kedir & Abduro (2019). The Cd concentrations in the muscle and liver tissues of both fish species were below the detection limit for the Omo river and Omo delta lake.

A T-test (p < 0.05) was conducted to compare the mean levels of heavy metals in the muscle and liver tissues of L. niloticus and N. niloticus. The results showed that there were significant differences in the mean levels of all heavy metals except for Mn, Cu, and Ni in L. niloticus and Zn and Co in O. niloticus. Due to their non-biodegradable and persistence nature in the environment, heavy metals cause toxicity in fish by producing oxygen reactive species through oxidizing radical production. Higher level of heavy metals in liver tissue in the present study may adversely affect fish physiology such as hemato-biochemical properties, immunological parameters especially hormones and enzymes, histopathology of different major organs (Shahjahan et al., 2022).

There were also species dependent significant differences in the mean levels of Pb, Ni, and Co in the liver tissues of L. niloticus and O. niloticus. Likewise, the mean contents of Pb, Fe, and Ni in muscle tissues of both species were significantly different (p < 0.05). Numerous researchers examined the possibility that variations in the accumulation of heavy metals across different species of fish could be linked to their habitat and eating preferences, such as whether they are omnivores, herbivores, or carnivores (Yilmaz et al., 2007; Elnabris, Muzyed & El-Ashgar, 2023). Variations in the mean concentrations of heavy metals between L. niloticus and O. niloticus in the current study may be attributed to variations in feeding habits and habitat use (Dadebo et al., 2014; Samuel et al., 2020). Biological factors including age and growing rates of fish species could also be attributing to differences in heavy metal concentrations between L. niloticus and O. niloticus (Yilmaz et al., 2007; Ahmed et al., 2015) of the present study.

The differences in the mean levels of heavy metals between the fish tissues (liver and muscle) in this finding could be due to the ability of various metals to bind with carboxylate oxygen, the amino functional groups, and nitrogen in metal-binding proteins (Uysal et al., 2009; Gure, Kedir & Abduro, 2019; Pramita et al., 2021). The variations between tissue levels of metals could also be ascribed due to differences in the physiological role of each tissue in which muscle generally accumulates lower levels of heavy metals (Ahmed et al., 2015; Olawusi, Ajibare & Akinboro, 2019). Many studies also confirmed that there was variation in heavy metal levels among fish tissues and species (Samuel et al., 2020; Gure, Kedir & Abduro, 2019), which was also observed in the current finding.

In general, L. niloticus showed a greater burden of heavy metals than O. niloticus. This could be due to differences in the behavior and feeding habits of the two species. Thus, the relatively high level of metals in the L. niloticus tissues in the present study could be attributed to their feeding habits as they are bottom-dwelling carnivores that feed on zooplankton, shrimp, clams, snails, insects and other fish species, unlike to O. niloticus which feeds on algae and other vegetables (Magu et al., 2016; Dadebo, Mengistou & Gebre-Mariam, 2005). Carnivores are more likely to accumulate heavy metals than other fish (Ahmed et al., 2015). Also, this finding has potential ecological implication particularly relating with the level of heavy metals in liver tissues of both fish species from the water bodies. The finding of the study revealed that liver tissue gad higher burden of heavy metals (Pradip et al., 2019). These can cause a variety of fish population morphology such as decrease of hatching rate, feeding behavior, reproductive system which in turn adversely affect aquatic bodies and aquatic ecosystems that has a significant influence on the food chain and freshwater ecology.

A strong significant positive correlation was observed between heavy metals Cr and Fe (r = 0.703) in the muscle of O. niloticus from both lower Omo river and Omo delta (r = 0.705). The heavy metals Fe and Ni (r = 0.65), and Co and Fe (r = 0.482) had moderate positive correlation whereas Fe and Zn (r = −0.23), and Cu and Zn (r = −0.19) weak negative correlation in the O. niloticus from lower Omo river. The strong positive correlation existed among the heavy metals could be due to the similar sources of pollution and similarities in behavior of heavy metals in the water bodies. The negative correlation existed among the heavy metals may due to the difference in source of pollution.

The THQs for heavy metals in fish muscle consumed by adults and children in all of the samples from the Omo river that were analyzed were less than one, indicating that people are unlikely to face significant health concerns as a result of ingesting a single heavy metal through consumption of the fish muscles. The HI of the discovered heavy metals was also less than one, indicating that, at the time of the study, there was no significant risk to human health from consuming L. niloticus and O. niloticus muscle tissues from the Lower Omo river source. The mean contribution of the THQ value to HI showed that Pb, Cu, and Mn contributed approximately 97% the HI through the muscle of fish tissues. Pb alone was responsible for 90% of the HI through the muscular tissues of the two fish species. Therefore, in regard to non-carcinogenic dangers, more attention should be given to the Pb level in the muscle of both fish species. Regarding the noncarcinogenic risks, Magna et al. (2021) reported that the THQ value for Mn (0.00325) was greater in their study conducted in the Volta Basin River, Ghana, than in the current findings for Mn (0.011). However, these authors reported lower THQs values for Ni (0.000108), Zn (9.2 × 10−5), and Fe (2.14 × 10−8) than the present study via intake of O. niloticus muscle by children. Similarly, Samuel et al. (2020) from their study in Ethiopia from Boicha stream (Hawassa) reported higher THQ values for Fe (0.01), Co (0.001), Ni (0.002), Cu (0.02), and Zn (0.039) than the present findings. However, they reported a lower THQ value for Pb (0.026) than was found in the present study (0.5368) for O. niloticus by an adult while consuming one to seven days a week.

For fish sampled from the Omo river, the target cancer risk (TCR) values in the muscle of both L. niloticus and O. niloticus were in the order of Ni > Pb. The TCRs values for Pb and Ni in this study were within the tolerable range of (10−6 to 10−4) (United States Enviromental Protection Agency (USEPA), 2012) for all levels of exposure. The highest TCRs were observed for nickel in L. niloticus and O. niloticus muscle consumed by children for Ni was 6.72 × 10−5 and 6.24 × 10−5 respectively. Similarly, the highest TCR value due to L. niloticus and O. niloticus by adults for Ni was 4.51 × 10−5 and 4.08 × 10−5 respectively. This demonstrated that, for all exposure levels, there was no risk to the health of L. niloticus and O. niloticus from ingesting Pb and Ni through muscle. It was also observed that children had a higher probability of developing risk when exposed to heavy metal pollution. Magna et al. (2021) from their study in Volta Basin River, Ghana, reported lower TCR for Ni in children (5.5 × 10−8) than the present study. Similarly, Samuel et al. (2020) from their study in Ethiopia from Boicha stream (Hawassa) reported higher TCR for Ni in adult (5.51 × 10−5) than the present study (4.08 × 10−5). However, they reported a lower TCR for Pb (7.65 × 10−8) than was found in the present study (1.58 × 10−5) via intake of O. niloticus muscle by adults.

The THQs for heavy metals in fish muscle consumed by adults and children was less than one in all of the Omo delta samples that were analysed, indicating that people are unlikely to face significant health hazards as a result of consuming fish muscles that contain heavy metals. The HI of the discovered heavy metals was also less than one, indicating that, at the time of the study, there was no significant risk to human health from consuming L. niloticus and O. niloticus muscle tissues from the lower Omo river source. As seen from the risk assessment data, more emphasis should be given to the carcinogenic risk of Pb in the muscles of both fish species. Magna et al. (2021), from their study in the Volta Basin River, Ghana, recorded a lower THQ value for Mn (0.00325) than the present findings for Mn (0.0062) from O. niloticus. They also reported lower THQ values for Ni (1.08 × 10−4), Zn (9.2 × 10−5), and Fe (2.14 × 10−8) than the present study via intake of O. niloticus muscle by adults and children. However, compared to the current results, Samuel et al. (2020) found that the THQ values for Fe (0.01), Co (0.001), Ni (0.002), Cu (0.02), and Zn (0.039) were greater in their study conducted in Ethiopia near Lake Hawassa. They did, however, disclose a lower THQ value for Pb (0.026) in muscle O. niloticus by an adult while consuming one to seven days a week than the current finding (0.384).

The target cancer risk (TCR) values in the muscles of both L. niloticus and O. niloticus were in the order of Ni > Cr > Pb in the Omo delta. The TCR values for Pb and Ni in this study were within the tolerable range of (10−6 to 10−4) (United States Enviromental Protection Agency (USEPA), 2012) for all levels of exposure. The highest TCRs were observed due to the consumption of L. niloticus and O. niloticus muscle was by children for Ni 1.11 × 10−4 and 7.58 × 10−5 respectively. Similarly, the highest TCR value due to L. niloticus and O. niloticus by adults for Ni was 7.75 × 10−5 and 5.3 × 10−5 respectively. This showed that, at all exposure levels, there was no risk of cancer from consuming Cr, Pb, or Ni through the muscle of L. niloticus and O. niloticus. It was also observed that children had a higher probability of developing risk when exposed to heavy metal pollution. Magna et al. (2021) from their study in Volta Basin River, Ghana, reported a lower TCR for Ni in children (5.5 × 10−8) than the present study (1.11 × 10−4). Samuel et al. (2020) from their study in Ethiopia from Lake Hawassa reported higher TCR for Ni in adult (5.51 × 10−5) than the present study (5.3 × 10−5). However, they reported a lower TCR for Pb (7.65 × 10−8) than the present study (1.22 × 10−5) via the intake of O. niloticus muscle by adults.

Conclusions

This study had the objective of measuring the levels of heavy metals present in the liver and muscle tissues of two commercially significant fish species, namely L. niloticus and O. niloticus found in the Omo river basin and Omo delta located in southern Ethiopia. The findings of the study provided the first baseline information on the level of nine heavy metals in these fish species from Ethiopian low land freshwater. The levels of all heavy metals evaluated, except of Pb, were within the permissible limits established by the FAO/WHO (1989) for the Omo river. Similarly the levels of all heavy metals under investigation, except Pb and Cr, were within the permissible limits established by (FAO/WHO, 1989) in the Omo delta. The liver and muscle tissues of L. niloticus were found to have higher accumulations of heavy metals than those of O. niloticus, with the liver accumulating more heavy metals than muscle tissues. Overall, the study suggested that there are possible risks to human health from heavy metals contamination in these fish species.

The level of heavy metal pollution found in fish tissue is a cause for concern. While the health risk assessments did not indicate any immediate danger to human health, the mean levels of Pb detected in the both liver and muscle tissue of two fish from the Omo river and Omo delta exceeded the allowed level set by FAO/WHO (1989). These suggests that regular monitoring of freshwater fish in this area is necessary. Furthermore, the TCR resulting from Ni exposure through the consumption of L. niloticus and O. niloticus muscle is alarming, as it may increase the risk of cancer in young people who engage in vigorous and prolonged developmental activities. Therefore, it is imperative to monitor heavy metal levels in the tissues of L. niloticus and O. niloticus, policy makers are advised to take appropriate action at this alarming level to safeguard freshwater fish and people from the threat of heavy metal pollution from the lower reaches of the river and Omo delta.

Supplemental Information

Supplemental Information 1 Raw Data.

Additional Information and Declarations

Competing Interests

Author Contributions

Ethics

Field Study Permissions

Data Availability

The authors declare that they have no competing interests.

Abiy Andemo Kotacho conceived and designed the experiments, performed the experiments, analyzed the data, prepared figures and/or tables, authored or reviewed drafts of the article, and approved the final draft.

Girma Tilahun Yimer conceived and designed the experiments, performed the experiments, prepared figures and/or tables, authored or reviewed drafts of the article, and approved the final draft.

Solomon Sorsa Sota conceived and designed the experiments, performed the experiments, prepared figures and/or tables, authored or reviewed drafts of the article, and approved the final draft.

Yohannes Seifu Berego conceived and designed the experiments, analyzed the data, prepared figures and/or tables, authored or reviewed drafts of the article, and approved the final draft.

The following information was supplied relating to ethical approvals (i.e., approving body and any reference numbers):

The Hawassa University granted Ethical approval to carry out the study within its facilities (Ethical Approval ref; Bio/499/13).

The following information was supplied relating to field study approvals (i.e., approving body and any reference numbers):

Field experiments were approved by Research council of Hawassa University (Bio/499/13).

The following information was supplied regarding data availability:

The raw measurement are available in Supplement File 1. The raw data shows the concetration of Heavy metals from two fish species and two water bodies.

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
