# Peer review of "Heavy metal levels and human health risk implications associated with fish consumption from the lower Omo river (Lotic) and Omo delta lake (Lentic), Ethiopia"

_PeerJ, doi:10.7717/peerj.17216_

## Round 0.1 · original submission · Major Revisions

The 3 reviewers emphasised the great importance of your study, but at the same time identified important points that require a fundamental revision of the manuscript before it can be accepted for publication. I fully agree with the assessment of the three reviewers and see a need for revision in the following areas in particular:

1) The paper needs a more sharpened focus, particularly in elucidating the potential sources and implications of the heavy metal contamination which would also open options for the management of the obvious environmental and human health problem for practitioners.

2) The effects of heavy metals on fish physiology should be discussed in more detail, including the role of liver contamination.

3) The presentation of the results should take into account a reorganisation of the tables, in which there are currently some redundancies. This would facilitate the interpretation of the results.

4) Regarding the statistical methods used, it should be examined whether multivariate methods such as principal component analysis (PCA) or correlation tests can identify possible interdependencies between the heavy metals. In any case, it should be checked whether the requirements for the Student t-test currently used in the manuscript as a parametric method are met at all (normal distribution and equality of variances). Otherwise, non-parametric methods should be used.

5) Although the experts have suggested some linguistic improvements, further optimisation is required.

I look forward to your revised manuscript.

**Language Note:** The review process has identified that the English language must be improved. PeerJ can provide language editing services - please contact us at copyediting@peerj.com for pricing (be sure to provide your manuscript number and title). Alternatively, you should make your own arrangements to improve the language quality and provide details in your response letter. – PeerJ Staff

·

Basic reporting

no comment

Experimental design

no comment

Validity of the findings

no comment

Additional comments

Dear Authors,

Peace be upon you and God's mercy.
Hoping you are fine.

Thanks for submitting your review “Analysis of Heavy Metals and Human Health Risk Implications Associated with Fish Consumption from the lower Omo River (Lotic) and Omo Delta Lake (Lentic), Semiarid Region of southern Ethiopia” to the PeerJ. I noticed that the current study aimed to assess human health risks of heavy metals in commonly consumed fish spices (L. niloticus and O. niloticus) collected from the lower Omo River (Lotic) and the Omo delta (Lentic). The concentrations of nine heavy metals (Cd, Co, Cr, Cu, Fe, Mn, Ni, Pb, and Zn) were measured in the samples, and the non-carcinogenic and carcinogenic health risks to adults and children associated with consuming fish were calculated respectively.
To Authors: Dear authors, first of all, thank you for this important and valuable study. The topic has been well-addressed. In order to improve your work further. I have seen a lot of problems in the whole review. My specific comments are listed below.
Keywords
Keywords are a tool to help indexers and search engines find relevant papers. If database search engines can find your journal manuscript, readers will be able to find it too. This will increase the number of people reading your manuscript, and likely lead to more citations. However, to be effective, Keywords must be chosen carefully. They should represent the content of your manuscript and be specific to your field or sub-field. Please, add the keywords after abstract section and capitalize the first letter of each word in the keywords.
Highlights
- Please, add the highlighted sentences.
- Please double-check the abbreviations in this manuscript.
Statistical analysis
This part is very important, but you need a biostatistician. There are many overlaps between which the differences must be clarified. Significant differences in different metals between the treatments (fish feeds) were not explained, so you need a biostatistician to determine the appropriate analysis to clarify the differences between the heavy levels between the two fish (L. niloticus and O. niloticus) and between two organs (muscles and liver), as well as the two areas (Omo River (Lotic) and Omo Delta Lake (Lentic)).
Results
The results are poorly presented in all tables. You can show the results in both tables and charts. You can also merge more than one table into one table such as the results of the t-test table merge it with the metal levels table. When calculating the non-carcinogenic risks (Estimated daily intake (EDI) and target hazard quotient (THQ)), the metal levels have been used as dry weight, and this is an error. Referring to previous studies, it was found that the metal levels used as a wet weight to calculate non-carcinogenic risks (EDI and THQ), and therefore you must recalculate this part. the weight of fish in wet was converted to the weight of fish in dry using a coefficient of conversion (4.8 as Rahman et al., 2012).
https://dx.doi.org/10.21608/ejabf.2022.244364.
DOI: 10.21608/ejabf.2023.314426.
https://doi.org/10.1007/s12011-023-03880-0.
References
- Finally, all literature cited in the text should appear in the final list and vice versa,
Kind regards

·

Basic reporting

The manuscript is a collection of wonderful writing but the authors must improve the manuscript before publishing. The authors should take into accounts the editorial and typographical errors in this manuscript and engage the service of a professional manuscript language editor.

Experimental design

The design of the experiment was fantastic based on the information provided for heavy metals analysis and health risk indices.

Validity of the findings

The authors were able to determine the THQ and HI for both adults and children. It shows that they take into account different class of man. It also show the levels of heavy metals in fish of lotic and lentic ecosystems.

Additional comments

Line 17. The study was done not A study was done
Line 17. Levels of heavy metals not level of heavy metals (same for other lines in the manuscript)
Line 18. Lates niloticus and Oreochromis niloticus not Lates niloticus and Oreochromis niloticus (italics should be removed)
Line 19. One hundred and twenty fish samples not (120 fish samples) and space between the parenthesis should be removed
Line 25. from lower Omo River…….. (Italics should be removed from this statement).
Line 26. whereas , space between whereas and , should be removed (whereas,)
Line 32. from the Omo delta…….. the statement should not be italicised
Line 34. and ….. should not be in italics
Line 36. and ….. should not be in italics
Line 36 FAO/WHO permissible (limits should be added to this statement).
Line 37. and ….. should not be in italics
Line 88. (L. niloticusandO. niloticus) should be (L. niloticus and O. niloticus)
Line 130. HNO3 should be HNO3
Line 165. (Neff et al.), there is no year for this citation
Line 175. (Gure et al.), there is no year for this citation
The authors should take note of necessary spaces after the end of some statements especially paragraph 2 of the Introduction.
Also, there should be spaces between the authors and year and where necessary (Line 63).
Also there should be space between numbers and its abbreviated unit (Line 127, 161).
Where necessary there should be spaces between some words (Lines 55, 57, 59, 62, 63, 65, 71, 72, 75, 97, 102, 104, 107, 112, 124, 138, 149, 150, 151, 164, 166, 169, 174, 188, 194, 221).
All statements related to table 1 should be together, ditto for other tables.
Note: All fish species name should be italicised.
Line 427. O. Niloticus should be written as O. niloticus
Probability (P<0.05) should be (p<0.05)
Line 458. Variations should be in lower case
Content associated with different metals in the manuscript should be changed to level, for example Cobalt content should be Cobalt levels in Line 441.
Line 441. Cobalt content in muscle tissues O. niloticus varied 0.054mg 442 kg-1 to 0.082 mg kg-1 in Omo River and Omo delta, should be written as Cobalt levels in the muscle tissues of O. niloticus varied from 0.054 mg to 0.082 mg kg-1 in Omo River and Omo delta respectively.
Line 445. Gureet al. (2019) should be written as Gure et al. (2019).
Lines 445 to 446. It was addressed by many researchers that fish species-dependent differences in heavy metal accumulation might be associated with feeding habits such as being carnivores, herbivores, or omnivores and habitat of fish species. The statement should be rephrase in a better dimension.
Line 460. ‘could also attribute to differences’…… should be ‘could also be attributed to differences’.
Lines 510, 525, 534, 536, 537. The exponential numbers should be in superscript.
 The authors did not mention the CPSo values of the heavy metals
 The map of the study area needs to be revised.
 The authors should provide more catchy figures related to HI and THQ as provided by https://doi.org/10.1016/j.heliyon.2023.e16609
 Information on the morphometric parameters of the fishes in the two waterbodies are not provided.

Reviewer 3 ·

Basic reporting

The manuscript offers a solid literature review and employs a rigorous methodological framework, particularly in the collection and analysis of samples using Flame Atomic Absorption Spectrometer (FAAS), which is commendable for its precision.
However, the presentation of the results needs to be more precise to enhance clarity and the overall impact of the study's findings. The paper falls short in contributing groundbreaking insights to the field, and there's a noticeable gap in the analysis of the origins and causes of heavy metal contamination. The discussion, while thorough in its review of the literature, does not sufficiently explore the physiological effects of heavy metals on fish, particularly regarding liver tissue. Moreover, the data could be more effectively presented by consolidating similar tables to reduce redundancy and improve the interpretability of results.

Experimental design

The analytical approach could be enhanced to offer more innovative insights into the distribution of heavy metals and their impacts. The study's generic analytical proposition could be strengthened by incorporating a Principal Component Analysis (PCA) to elucidate the distribution of heavy metals and the effects of anthropogenic activities in the sampled areas. This approach would not only reinforce the findings but also provide a more nuanced understanding of heavy metal dispersion. Furthermore, while bioaccumulation coefficients are analyzed, a more in-depth physiological study could elucidate the behavior of these metals within the fish's body, particularly concerning the liver's role in metal assimilation. Implementing PCA could also aid in examining the relationships and correlations between different heavy metals. Regarding statistical analysis, while the use of Student's t-test is apt for comparing standard limits, a sectoral comparative analysis might offer a clearer picture of significant differences between different river sections, enhancing the study's granularity and practical implications.

Validity of the findings

The findings of the study are of significant relevance given the public health implications associated with heavy metal contamination in fish, a critical issue in population health. The data presented are robust, encompassing not only the commonly consumed muscle tissue but also the liver, which offers a more comprehensive understanding of bioaccumulation. However, the data analysis and presentation could benefit from enhancements to avoid redundancy and improve statistical robustness. While the article employs a Student's t-test, it does not provide a clear rationale for this choice in the methodology, nor does it confirm if the assumptions of normality and equal variances are met. It would be advisable to consider whether non-parametric statistical tests could be more appropriate given the data distribution.

Further, the study would gain substantial analytical depth from advanced statistical approaches such as Principal Component Analysis (PCA) or correlation tests, which could provide greater insights into the patterns of heavy metal distribution and the potential relationships between different metals within the fish tissues. Although the conclusions are well articulated and linked to the research question, they should strictly reflect the results supported by robust statistical evidence. Thus, to improve the validity of the findings, a critical assessment of statistical methodologies is recommended, ensuring that the results can be confidently interpreted and contribute valuable knowledge to the existing literature on environmental health risks.

Additional comments

For additional refinement, the authors should consider including spatial analysis to better elucidate the distribution of heavy metals across the study sites, which could offer vital environmental insights. It is also recommended to incorporate a discussion on the potential ecological implications of the findings, linking them to broader environmental health contexts.

Annotated reviews are not available for download in order to protect the identity of reviewers who chose to remain anonymous.

---

## Round 0.2 · Minor Revisions

We are almost there. A few editorial aspects as raised by reviewer 2 should be resolved before the final acceptance of your paper.

·

Basic reporting

Dear Authors,

I am writing to express my approval for the publication of the research titled "Heavy metal levels and human health risk implications associated with fish consumption from the Lower Omo River (Lotic) and Omo Delta Lake (Lentic), Ethiopia." The study's findings regarding the levels of heavy metals in fish tissue samples and the associated human health risks are of significant importance for environmental and public health concerns.

I have thoroughly reviewed the study and the authors' responses to the comments. The revisions made address the concerns raised by the reviewers, and I believe that the manuscript is now suitable for publication.

Thank you for your valuable contribution to this important field of research.

Best regards

Mahmoud Mahrous M. Abbas

Experimental design

Good

Validity of the findings

Very Good

Additional comments

No

·

Basic reporting

The literature review is up-to-date. Although, the manuscript still needs improvement from the authors. Also, the authors should take note of the editorial and typographical errors in the manuscript, and I still maintain that the authors should engage the services of a manuscript editor.

Experimental design

The design of the experiment was fantastic based on the information provided for heavy metals analysis and health risk indices.

Validity of the findings

No comment

Additional comments

Line 17. Lates niloticus and Oreochromis niloticus (it should be italicised). Sorry for the error.
Line 27. respectively.Similarly,the…… should be respectively. Similarly, the
Line 33. O.niloticus should be O. niloticus
Line 35. There should be space between full stop and The Pb level
Line 38. fish.Consequently,……….. should be fish. Consequently,
Line 60. The author need to put space between full stop and Heavy metals
Line 62. There should be space between full stop and The danger
Line 64. There should be space between full stop and Humans
The authors should take note of necessary spaces after the end of some statements especially paragraph 3 of the Introduction.
Where necessary there should be spaces between some words (Lines 66, 67, 70, 77, 107, 108, 110, 115,121, 141, 150, 151, 167, 186, 192, 220, 226, 259, 307, 317, 333, 370).
 The authors did not mention the CPSo values of the heavy metals
There should be spaces between numbers and their unit.

Reviewer 3 ·

Basic reporting

I am pleased to note the improvements in your manuscript, particularly in the clearer presentation of results and better organization of the methods section. These changes have significantly enhanced the readability and coherence of your work.

Experimental design

While the multivariate analysis improvements are partly correct, I recommend incorporating more robust multivariate techniques in future work. Nonetheless, the enhanced presentation of your current analysis is a notable improvement.

Validity of the findings

The article excels in establishing the validity of its results, which are poised to significantly aid in understanding and improving health policies related to heavy metals in ichthyological resources like fish. This contribution is both timely and valuable.

Additional comments

I recommend the publication of this article, considering the author has effectively implemented the suggested changes, enhancing the overall quality and impact of the work

---

## Round 0.3 · Minor Revisions

Thank you for the revision of the manuscript. Although you have adequately taken into account the content-related reviewers' comments, there are still many issues with English grammar and also with the formatting. I strongly advise to sent your manuscript to a language editing service (also provided by PeerJ, see https://peerj.com/pricing#editing-services) reviewed by a colleague to improve the readability.

**Language Note:** The Academic Editor has identified that the English language needs to be improved. When you prepare your next revision, please either (i) have a colleague who is proficient in English and familiar with the subject matter review your manuscript, or (ii) contact a professional editing service to review your manuscript. PeerJ can provide language editing services - you can contact us at copyediting@peerj.com for pricing (be sure to provide your manuscript number and title). – PeerJ Staff

---

## Round 0.4 · Minor Revisions

The quality is better than in the previous version but still far away of being good. There are also still numerous slips of the pen such as missing blanks between words. I have started to edit the abstract and first paragraph of the introduction but stopped because there were so many changes necessary (see attachment).

I strongly recommend to use a professional service to improve the language to meet our standard of using professional English before your submission can be accepted.

**Language Note:** The Academic Editor has identified that the English language must be improved. PeerJ can provide language editing services - please contact us at copyediting@peerj.com for pricing (be sure to provide your manuscript number and title). Alternatively, you should make your own arrangements to improve the language quality and provide details in your response letter. – PeerJ Staff

---

## Round 0.5 · accepted · Accept

Thank you for the revision of the manuscript. I hereby certify that you have adequately taken into account the reviewers' comments and also improved the language issues. There are still a few slips of the pen such as missing or unnecessary blanks between words, but this can be resolved during the correction of the proofs. as I have checked by my own assessment of your revised manuscript. Based on my assessment as an Academic Editor, your manuscript is now ready for publication.